# Is There a Correlation between Dental Occlusion, Postural Stability and Selected Gait Parameters in Adults?

**DOI:** 10.3390/ijerph20021652

**Published:** 2023-01-16

**Authors:** Monika Nowak, Joanna Golec, Aneta Wieczorek, Piotr Golec

**Affiliations:** 1Faculty of Medicine and Health Sciences, Andrzej Frycz Modrzewski Krakow University, 30-705 Kraków, Poland; 2Institute of Clinical Rehabilitation, University School of Physical Education in Krakow, 31-571 Kraków, Poland; 3Department of Prosthodontics and Orthodontics, Faculty of Medicine, Jagiellonian University Medical College, 31-007 Kraków, Poland; 4Individual Medical Practice, 30-390 Kraków, Poland

**Keywords:** malocclusion, postural stability, CoP, gait

## Abstract

Background: There is still an ongoing debate about the role of the craniomandibular system, including occlusal conditions, on postural stability. This study aims to assess the role of antero-posterior malocclusion on postural control and plantar pressure distribution during standing and walking. Methods: 90 healthy volunteers (aged 19 to 35) were qualified for the study. The subjects were assigned to three groups, depending on the occlusion type. Each group (Angle Class I, II and III) consisted of 30 people. The research procedure included a clinical occlusal assessment performed by a dentist. Postural control measurements were carried out using a force platform by measuring plantar pressure distribution during standing (six trials with and without visual control) and walking test conditions. Results: The tendency to shift the CoP forward is demonstrated by Angle Class II subjects and backwards by Class I and III subjects (*p* < 0.001). Individuals with a malocclusion demonstrated significantly higher selected stabilographic parameters while standing on both feet (with eyes open and closed) and during the single-leg test with eyes open (*p* < 0.05). The analysis of the dynamic test results showed no significant correlations between Angle Classes and the selected gait parameters. Conclusions: Analyses conducted among individuals with malocclusions showed the impact of occlusion on static postural stability. In order to diagnose and effectively treat malocclusion, a multidisciplinary approach with the participation of dentists and physiotherapy specialists is necessary, with the use of stabilometric and kinematic posture assessments.

## 1. Introduction

Posture is understood as the position of the human body and its orientation in space. The development of a person’s posture is individual and largely dependent on the myofascial and skeletal structure and function. Maintaining a stable standing position is possible through precise neuromuscular coordination of all body segments. It requires analysing and integrating stimuli from three systems: vision, vestibules and proprioception [1,2]. Numerous observations have been made over the years on factors influencing postural stability [3,4,5,6,7]. The role of the craniomandibular system is now increasingly being analysed in connection with it [8,9,10,11]. Many theories attempt to explain the association between the masticatory organ and posture, including myofascial chains, activation or deactivation of the trigeminal nerve and subsequent interaction in the brainstem [12,13,14]. However, this is a contentious topic in the scientific community. There is evidence both supporting such a relationship [15,16,17,18,19,20] and refuting it [21,22,23,24].

The authors of scientific reports, who recognise the associations between the systems in question, give two directions for possible interactions. The first one, i.e., ascending disorders, refers to the situation in which poor posture and disorders in the peripheral structures (e.g., lower limbs), through the myofascial trains and the dura mater, affect the craniomandibular structures. In contrast, a chain of descending disorders is present when abnormalities in the craniomandibular region affect posture and areas of the body that are located more distally, also involving the pelvis and lower limbs [12,25,26,27]. Previous publications attempting to assess the impact of craniofacial abnormalities on posture and stability focus, among others, on the analysis of patients presenting symptoms of temporomandibular disorders. It has been shown that changes in the temporomandibular joint (TMJ) can directly impact muscle activity in terms of posture, stability and physical performance [16,28,29]. Positive effects of occlusal splints on posture in patients with temporomandibular disorders (TMD) have also been found. However, the lack of high-quality studies using advanced measurement tools to understand better the phenomenon under study is highlighted [30].

The present study attempts to evaluate the impact of masticatory abnormalities on postural control and focuses on assessing individuals with specific malocclusions that determine the anteroposterior position of the mandible. According to some researchers, malocclusion, such as TMD, can affect the osteoarticular system of the whole body and become a source of persistent pain and promote the development and perpetuation of certain postural defects. According to the authors cited, occlusal disorders can result in altered stimulation of periodontal proprioceptors, causing changes in the tension of neck muscles and postural muscles and changes in the position of the head, followed by compensatory changes in the anatomical regions in their immediate vicinity. Over time, this can affect posture, the position of the centre of gravity or foot contact with the ground [12,25,26,31,32]. However, there is still a gap in scientific knowledge on the relationship between craniofacial structure and spinal postural control in patients with malocclusion. In addition, the available papers show problems related to the small number of subjects, the small number of parameters tested or the selection of reliable measurement tools [22,33,34].

Malocclusion, on which these studies focus, can result from abnormalities in the structure and alignment of the bones of the maxilla and mandible in relation to each other or from an abnormal arrangement of dental arches. Angle suggested a classification of occlusion and malocclusion based on the anteroposterior position of the first molar and the position of canines [35,36]. Malocclusion is often a congenital condition, resulting from hereditary or environmental factors. It is also caused by local factors, such as an abnormal pattern of breathing or postural defects, as well as oral parafunctions such as nail biting or teeth grinding (bruxism) [36]. According to analyses by Lombardo, occlusal abnormalities occur on average in 56% of the general population [37]. Their prevalence increases with age. Given the increase in their prevalence in subsequent age groups and the consequences they entail, it is reasonable to expect a large number of adult patients who will need complex and expensive multidisciplinary treatment [37,38].

Given the high proportion of patients with malocclusion [19,20] and the contradictory reports concerning the relationships in question [15,16,17,18,19,20,21,22,23,24], the need for further knowledge and analysis of individual malocclusions and accompanying musculoskeletal abnormalities in dynamic and static conditions is reasonable. There is still a lack of research regarding the effect of occlusion on postural stability and plantar pressure distribution during standing and gait in the same group of adults with Angle Class I, II and III. Therefore, this is the main objective of this study. Both standing and gait are primary forms of human motor activity, so both motor tasks were analysed. Moreover, this study was extended to include the analysis of variables during the performance of more difficult motor tasks (single-leg stance) and in the presence and absence of visual control. The development of diagnostic methods creates more and more opportunities to undertake research in this area. Computer-based systems and dynamometric platforms help obtain integrated and standardised data to measure the components of ground reaction forces during standing and gait and assess postural stability. Measuring the plantar pressure distribution during standing and gait is thought to have the potential to provide important information on postural control and, more specifically, body oscillation during standing. This makes it possible to prepare an appropriate therapy and eliminate the deficits that have occurred [39,40,41,42].

Given the gap in scientific knowledge, we believe that a better understanding of the relationship between craniofacial structures and postural stability in patients with occlusal conditions is needed.

Therefore, this study aims to evaluate the effect of anteroposterior malocclusion on postural stability and plantar pressure distribution during standing and gait in adults.

## 2. Materials and Methods

The study was conducted in the Biomechanics Laboratory of the Faculty of Medicine and Health Sciences at the Andrzej Frycz Modrzewski Krakow University from April 2020 to August 2021. The study involved Caucasian volunteers from the Lesser Poland Voivodeship, aged 19 to 35—patients of Krakow dentist’s surgery and Krakow university students. Eligibility criteria for inclusion in the study included the following: age between 18 and 35 years; at least 28 permanent teeth; Angle Class I, II or III occlusion. Exclusion criteria included braces or neurological, cardiovascular, osteoarticular or eye conditions that could result in gait and balance disorders. A total of 420 individuals were screened, and after considering the relevant inclusion and exclusion criteria, 90 individuals—52 women (57.78%) and 38 men (42.22%)—were included in the study.

The subjects were assigned to groups according to the occlusion type. The first group consisted of 30 individuals—19 women and 11 men (mean age: 22.77 ± 2.24 years) diagnosed with Angle Class III, or anterior occlusion, according to a dental examination. The second group consisted of 30 individuals—16 women and 14 men (mean age: 23.87 ± 3.9 years), diagnosed with Angle Class II, i.e., distal occlusion. The third group consisted of 30 individuals—17 women and 13 men (mean age: 22.63 ± 2.65 years)—diagnosed with normal occlusion (Angle Class I).

The study was approved by the Ethics Committee of the District Medical Chamber in Krakow (reference number 35/KBL/OIL/2019 of 19 February, 2019).

The research procedure included a clinical occlusal assessment and a functional examination of the masticatory organ based on the RDC/TMD protocol (Axis I and Axis II). The same dentist, a specialist in dental prosthetics, trained and calibrated in the RDC/TMD procedure, carried out the examination. In addition, based on an extraoral and intraoral examination, the anteroposterior relationships of the maxilla and mandible were evaluated using the Angle Classification, as mentioned above.

Stabilography and analysis of the ground reaction forces during standing and gait were performed using a baropodometric platform (FreeMED Base, manufactured in Italy) and the FreeStep 2.0 software (Figure 1). The platform consisted of an active (40 × 40 cm) and passive path (2 × 100 cm), with sampling frequencies up to 400 Hz. The measurement results were presented as a software-generated report containing information on the assessed parameters. The subjects were instructed not to consume alcohol the day before and on the day of the measurements, not to perform intense physical exercise, to stay well hydrated and to get plenty of sleep. The examinations were conducted in a closed, well-lit room with a temperature of 21 to 22 °C.

During the static measurement, patients were asked to stand barefoot on the platform and assume a relaxed position, with upper limbs lowered along the body and feet hip-width apart. Then they were asked to fix their gaze on a point on the opposite wall. During individual trials, patients’ bite was in the habitual position. The static test assessed such parameters as forefoot and hindfoot weight-bearing [%], mean forefoot and hindfoot pressure [g/cm^2^], centre of pressure (CoP) displacement in sagittal and frontal planes and lateral CoP displacement [cm].

Stabilography involved recording and analysing CoP displacements while maintaining balance on a single-plate platform—the FreeMED Base. The CoP projection of the feet on the ground was recorded as a point and a dynamic parameter, changing its position per unit of time. The study results are presented as a statokinesiogram, a stabilogram and numerical data. Stabilography consisted of six tests that required maintaining balance on a platform in the following standing positions:(1)Standing on both feet with eyes open;(2)Standing on both feet with eyes closed;(3)Standing on the left foot with eyes open;(4)Standing on the right foot with eyes open;(5)Standing on the left foot with eyes closed;(6)Standing on the right foot with eyes closed.

During the first two trials, the subjects were asked to maintain a given position for 30 s. Trials included three to six tests to assess postural stability while standing on one foot (with eyes open or closed). Correct positioning on the platform required a single-leg position stance (standing on the foot under test), bending the non-tested lower limb 90° at the knee joint and positioning the upper limbs along the body. The subjects were required to maintain a given position for 10 s. There was a five-minute break between individual tests, during which the subjects rested and sat down. The stabilography measured the following parameters: ellipse area [mm^2^], length (sway area) of statokinesiograms defined by CoP [mm] and mean CoP sway rate [mm/s]. The Romberg ratio (RR), the quotient of the test performed with eyes closed to the same test with eyes open, was also calculated. The higher the RR value, the greater the effect of the visual stimulus on postural control. An RR of more than 1 was considered significant [43]. The RR was calculated for both bipedal and single-leg stance tests for the following parameters: CoP path length, ellipse area and mean CoP sway rate.

The FreeMed Base platform was used for gait measurements during the dynamic test. Patients were asked to walk the path (240 cm long) 10 times, starting from the right foot, at a moderate pace, with the habitual bite position. The gait analysis included the assessment of the following parameters: length of gait line [mm], distribution of foot weight-bearing including forefoot, hindfoot, lateral foot and midfoot [%], mean foot movement rate [mm/s], mean foot pressure [kg].

Statistical calculations were performed using the R software, version 4.1.1 [44]. The significance level was *p* < 0.05. The quantitative variables were compared in the three groups using the Kruskal–Wallis test. When statistically significant differences were detected, the post hoc analysis with Dunn’s test was used to identify statistically significantly distinct groups.

## 3. Results

The analysis of the age and body structure of subjects with different occlusal types (Angle Class I, II and III) showed that both age and body proportions in these groups were similar. Basic data describing each group are shown in Table 1.

The analysis of the correlation between the type of dental occlusion according to the Angle Class and the foot load distribution in the statics condition showed significant differences between the studied groups (*p* < 0.05) (Table 2). Significantly higher values of the percentage weight-bearing [%] and the mean value of pressure distribution [g/cm^2^] on the left and right forefoot were noted in people with Angle Class II malocclusion compared to those with Angle Class I and III. On the other hand, people with Angle Class III and Class I had significantly higher values of the percentage weight-bearing of the hindfoot (*p* < 0.05). The mean pressure values in the hindfoot [g/cm^2^] of the lower right limb was significantly greater in people with Angle Class III than those with Class I and II (Table 2).

The analysed parameter of the static test was the CoP displacement, indicating its position in relation to the centre of the support polygon. The analysis of the correlation between the type of occlusion and the CoP displacement during the statics test showed significant differences between the studied groups (Table 3). The CoP displacement in the sagittal plane in patients with Angle Class II malocclusion was mostly forward, and in patients with Angle Class I and III, it was backward (*p* < 0.001) (Table 3). The results of the displacement and direction of CoP displacement in the frontal plane showed no significant differences between subjects with Angle Class I, II and III (*p* > 0.05) (Table 3). However, the size of CoP lateral displacements [cm] was significantly greater in subjects with the Angle Class III, compared to those with Class I (*p* = 0.009).

In the stabilographic analysis, in the test on both feet with eyes open, the ellipse area [mm^2^] and the CoP path length [mm] were significantly greater (*p* < 0.05) in people with malocclusion (Angle Class II and III) compared to those with Angle Class I (Table 4). In addition, the mean CoP sway rate [mm/s] was significantly higher (*p* < 0.05) in subjects with Angle Class III, compared to those with Angle Class I (Table 4). In the test on both feet with eyes closed, there were also significant differences between the groups (*p* < 0.05). The CoP path length [mm] and the mean CoP sway rate [mm/s] were significantly higher (*p* < 0.05) in subjects with Angle Class III compared to those with Angle Class I. Significantly higher values of the ellipse area [mm^2^] were noted in people with malocclusion (Angle Class II and III) compared to those with Angle Class I (Table 4).

The single-leg stance test with eyes open was performed first on the left lower limb and then on the right. There were significant differences between the study groups in terms of the stabilographic examination results but only in relation to the left lower limb (*p* < 0.05) (Table 5). The CoP path length [mm] and the mean CoP sway rate [mm/s] during standing on the left foot with eyes open were significantly greater (*p* < 0.05) in people with Angle Class III than those with Angle Class I (Table 5).

The analysis of the results of the single-leg stance test with eyes closed showed no significant differences between the analysed groups in the assessed parameters describing the postural stability of the subjects (*p* < 0.05) (Table 6).

Further analysis determined the Romberg ratio (RR), i.e., the ratio of the parameters in posturographic tests carried out with the eyes closed to the tests carried out with the eyes open. The correlation analysis showed no significant differences between the studied groups in terms of RR for the assessed parameters (*p* > 0.05) (Table 7).

The analysis of the dynamic test results showed no significant correlations between the occlusion type and the selected gait parameters (*p* > 0.05), i.e., the length of the gait line [mm]; the distribution of the foot’s weight-bearing including forefoot, hindfoot, lateral foot and midfoot [%]; the mean foot propulsion rate [mm/s]; and the mean foot pressure [kg] (Table 8).

## 4. Discussion

Despite the growing number of studies correlating malocclusion with postural disorders and postural stability, most show some difficulties and limitations. Predominant difficulties and limitations include small or unequal study groups, lack of a control group, incomplete descriptions of samples, a small number of parameters tested and the choice of reliable measurement tools [22,33,34,45]. Furthermore, given the clinical impact that the correlation between malocclusion and musculoskeletal disorders can have and the conflicting information on this subject, extensive scientific research is needed to prove and further understand this problem.

The foot is a functional unit stabilising the rest of the locomotor system and is the first link in the kinematic chain. It is also the first receiver and transmitter of impact, tension and compression. Each foot should be evenly weighted when standing, and the ratio of the forefoot to hindfoot pressure distribution should be approximately 4:6 when the feet are parallel to each other [46,47]. The interdisciplinary approach to treating patients with malocclusion resulted in questions as to whether abnormalities within the stomatognathic system can affect such a remote area as the feet, which support the human body, and, if so, what form this reciprocal relationship may take. This study also attempted to evaluate the relationship between the stomatognathic system and the biomechanics of the feet. Using a platform for assessing ground reaction forces, selected static test parameters were evaluated, mainly those related to weight-bearing distribution and the position of the CoG of a human body in individuals with different occlusion types.

Cuccia and Caradonna (2009) conducted a study using a pedobarographic platform and assessed the footprints of 84 individuals with TMD and 84 healthy individuals. They found differences in the plantar arch between these two groups and observed changes in forefoot and hindfoot weight-bearing distribution after artificially induced occlusal imbalance [18]. Cabrera-Domínguez et al. (2021) and Pérez-Belloso et al. (2020) found statistically significant differences between the foot contact area and the CoG of a human body in different occlusal classes. They noted a forward shift of the CoG in Angle Class II patients and a backward shift of the CoP in Angle Class I and III patients [10,12]. The results of this study are consistent with the above-mentioned observations [10,12]. This study also found that Angle Class II patients had their body CoG significantly more often shifted forward, while those diagnosed with Angle Class I and III had it shifted backward (*p* < 0.001). Consequently, the static test values for forefoot percentage weight-bearing and mean forefoot pressure distribution [g/cm^2^] were also significantly higher in Angle Class II patients, while the hindfoot values were significantly higher in Angle Class III and I patients. According to some researchers, such a shift of the CoG of the human body in Angle Class II and III patients can be explained by the fact that Angle Class II means a retracted mandible, which, in order to improve upper airway patency and compensatory tension changes in the upper quadrant of the body, can result in the head moving forward and thus shifting the CoG of the body forward. In Angle Class III, the reverse mechanism is present—the head is positioned in the posterior plane and the CoP is shifted backward [48].

An important part of the conducted analyses was the study of postural stability in subjects with different occlusal classes to test the possible relationship between the presence of malocclusion and postural control disorders.

Many researchers analysed various factors that can affect posture and its stability. In doing so, the impact of head and neck positioning, mood state, anxiety, breathing pattern, visual system and inner ear function was demonstrated [49,50,51,52]. There were also attempts to determine the impact of the stomatognathic system on postural control. In this case, however, many results confirming and refuting this impact are observed, including many studies with a small number of parameters tested [15,16,17,20,22,33]. Most studies report on the impact of artificially induced occlusal changes on the postural stability of subjects [42]. Slightly fewer studies address skeletal malocclusion and its impact on postural control. Available analyses primarily assess postural control in standing with both feet or on balance platforms, with eyes open and closed [53,54]. This study was enriched by considering the muscular system’s ability to maintain a stable body position while standing on a single leg. This also made it possible to assess neuromuscular coordination in subjects with different types of occlusion with a more difficult motor task.

According to some authors, the vestibular system regulates body balance-related processes based on information from three main postural sensors: the feet, the eyes and the masticatory organ. If there is an abnormality in one of these sensors, it will have an impact in the form of a disruption of information transmitted and posture. Therefore, proprioceptive afference from occlusion plays a vital role in maintaining proper postural control [55]. The impact of periodontal receptors on posture was analysed, among others, by Gangloff et al. (2000) who found a significant change in postural control after unilateral mandibular nerve anaesthesia, resulting in a shift of weight to the opposite limb and a worsening of the posturographic parameters analysed [56]. On the other hand, Sforza et al. (2006) found that improvements in the symmetry of the mandibular positioning resulted in a more symmetrical contraction of the sternocleidomastoid muscles and a reduced postural sway [32].

This study found differences between groups in terms of the posturographic parameters selected to assess the performance of the postural control system in subjects. It found that in a bipedal-stance test with eyes open, both the path length of the CoP sway [mm] and the ellipse area [mm^2^] were significantly greater in subjects with malocclusion (Angle Class III and II) compared to those with normal occlusion. However, the highest mean values were observed in subjects with anterior occlusion. The mean CoP displacement rate [mm/s] was also significantly higher in those with anterior occlusion compared to those with normal occlusion (*p* < 0.05). The results of this study are in line with recent observations by Ohlendorf et al. (2021), who also found that malocclusion, specifically Angle Class II, has a negative impact on postural control. Ohlendorf et al. observed that the greater the severity of the Class II malocclusion, the greater the ellipse area and its height on posturographic examination [53]. However, it should be emphasised that this study analysed young adults, while Ohlendorf et al. studied women aged 41–50 years, and, as Szczygieł et al. (2012) report, age and sex can also have an impact on postural control [53,57]. As considered by Arumugam (2016), patients with skeletal malocclusion also show reduced stability and increased postural sways as the severity of the malocclusion increases [58]. Álvarez Solano et al. (2020) in their review, reported changes in postural control and CoP displacement in terms of malocclusion, mainly in the anteroposterior direction, which was also observed in this study [59]. Julià-Sánchez et al. (2015) also suggested a link between the stomatognathic system and postural control, encouraging the investigation of possible postural changes during orthodontic treatment [54].

In addition to the Romberg test, this study also used single-leg stance tests to assess possible differences between the different occlusion classes in terms of the capacity of the postural control system for a more difficult motor task. It found that those with Angle Class III had significantly higher CoP path length values and sway rates in both single-leg stance and bipedal-stance tests. Higher mean CoP sway rates can be interpreted as an increase in the frequency of postural corrections made by those with malocclusion to maintain postural balance. These tendencies only change when two factors overlap: the performance of a more difficult motor task and a lack of visual control.

Visual information allows us to spatially determine the body’s position in relation to surrounding objects. Its absence results in the activation of postural control mechanisms associated with increased postural sway, which, according to some researchers, can significantly affect stabilometric parameters [60]. This study involved a posturographic analysis both in the presence and absence of visual control. The bipedal-stance test with eyes closed revealed similar tendencies compared to the same test with eyes open. The CoP path length [mm] and mean CoP displacement rate [mm/s] were significantly greater in Angle Class III patients than in Angle Class I patients. The single-leg stance test with eyes closed revealed no significant differences between the study groups and the parameters that describe postural control. Therefore, it may be assumed that the overlap of two factors, i.e., a more difficult motor task and a lack of visual control, created very unstable conditions for the system which seriously impede the maintenance of a stable posture, regardless of the occlusion type. This situation required the inclusion of additional balance maintenance mechanisms associated with increased postural sway. Interestingly, no effect of occlusion on postural control was demonstrated under such unstable conditions. Significant differences between classes became apparent when only one of the specified conditions occurred, i.e., a more difficult motor task (single-leg stance test) but with eyes open or an easier motor task (bipedal-stance test) without visual control. Observations of this study are in contrast to the results obtained by Tardieu et al. (2009) who argue that occlusal conditions can only impair postural control under unstable conditions, including those induced by the balance platform and in the absence of visual control [8].

The number of studies that analyse possible associations between the stomatognathic system and the organ of movement in relation to locomotion is small [42,61,62,63,64]. However, the importance of locomotion in daily life and the lack of current analyses that address skeletal malocclusion made the assessment of the relationship between anteroposterior malocclusion and gait parameters one of the aims of this study. Moreover, to the best of our knowledge, there are no available studies conducted among the same group of adults with different Angle Classes that assess parameters related to weight-bearing distribution and foot pressure forces both during standing and gait.

Available observations indicate that artificially induced changes in occlusal conditions can have immediate effects and affect locomotion, as demonstrated by the analysis of foot weight-bearing during gait in the study by Tecco (2010), as well as during running, as reported by Maurer et al. (2015) [61,65]. Furthermore, Fujimoto (2001) and Miles (2004) also observed that muscle activity and gait speed depend on the position of the craniomandibular system [62,66]. On the other hand, Maurer (2021)—similarly to this study—used a ground reaction force platform to determine the effect of different occlusion conditions on the distribution of foot weight-bearing during standing and gait. Maurer found no differences between the seven states of occlusion and pressure distribution during standing and gait. This study also revealed no significant differences between the analysed parameters of the dynamic test and the type of occlusion (*p* > 0.05). Moreover, the analysis of the results leads to the conclusion that the measurements of pressure distribution during standing and gait are independent of each other, which coincides with the reports by Maurer (2021) [42].

Studies by Fujimoto (2001) and Tecco (2010), in which occlusal abnormalities and immediate changes in gait parameters were artificially induced, showed a mechanism for the interaction of myofascial chains or activation and deactivation of the trigeminal nerve and its subsequent interactions in the brainstem [13,14,61,62,65,67,68]. However, in contrast to the authors cited [61,62], this study did not artificially induce occlusal changes. Instead, it focused on the assessment of gait parameters in those with three different occlusion types which represent a morphological feature of the patient—to which the body may have had the opportunity to adapt and produce for years some compensatory mechanisms to allow the most stable and ergonomic way of moving.

An increasing number of scientific reports on the possible association and co-occurrence of malocclusion with postural disorders have led dentists to increasingly recognise the need for cooperation and physiotherapeutic intervention in assessing postural disorders that could affect or impede satisfactory orthodontic treatment outcomes. Therefore, physiotherapists are essential to the interdisciplinary team when treating TMD patients and orthodontic patients. Specialists involved in orthopaedic physiotherapy, in general, are also recommended to take the stomatognathic system into account during a comprehensive evaluation of the patient’s locomotive organ, as abnormalities in this area involving myofascial chains can affect distal structures, affecting posture and its stability.

Research in physiotherapy should meet the requirements and standards of evidence-based medicine [69], hence the importance of using modern and objective measurement tools. The results of this study may suggest some tendencies in those with malocclusion, both in terms of changes in postural stability and the static distribution of foot weight-bearing. However, it should be emphasised that these analyses do not and cannot explain the causality of the observed associations. Given the still small number of reports and their contradictory findings, there is a need for further analyses to reveal and understand better the links between the knowledge areas discussed. Although this paper provides important data on the relationship between Angle Classes and selected static and dynamic test parameters on the force platform, which in this combination represents a gap in available research, perhaps a more detailed analysis using electromyography or advanced motion analysis systems is necessary to determine the relationships of these variables better.

## 5. Conclusions

Based on the results of the studies, it can be concluded that there is a clear relationship between occlusion and static balance. Patients with anteroposterior malocclusion show worse postural stability and a tendency to shift their centre of gravity.

A multidisciplinary approach involving specialists in dentistry and physiotherapy is essential for the proper diagnosis and effective treatment of malocclusion. The combined diagnostic strategy should consist of an occlusal assessment along with a stabilometric assessment of posture. Analyses conducted among patients with malocclusions emphasise the importance of correct occlusion in maintaining functional balance throughout the body.

No effect of malocclusion on the biomechanical characteristics of gait was demonstrated, suggesting that the measurements of plantar pressure distribution during standing and walking are independent.

## Figures and Tables

**Figure 1 ijerph-20-01652-f001:**
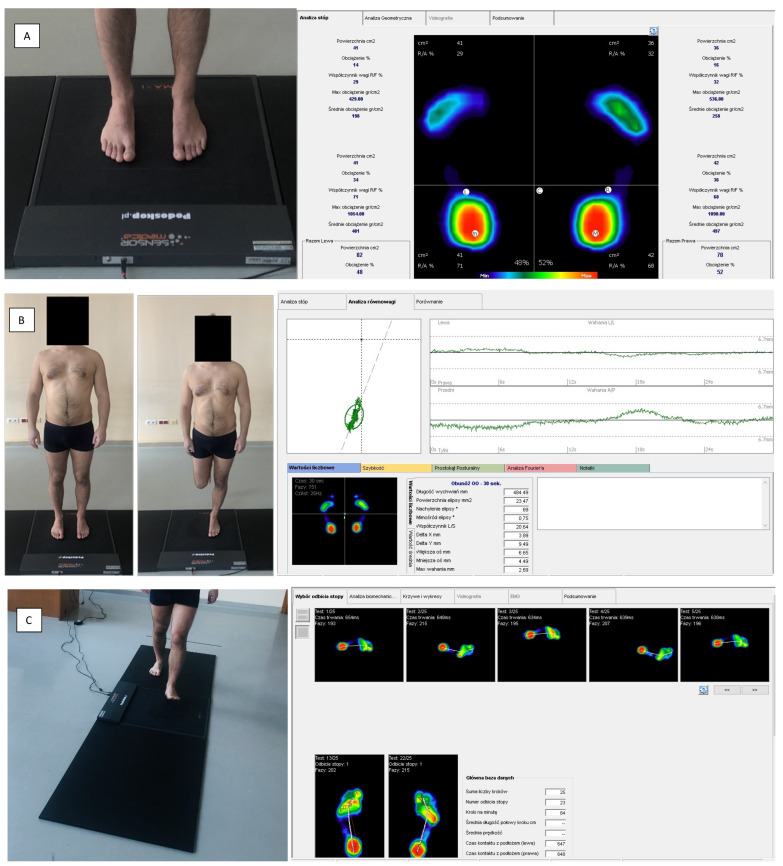
Measurement stand, equipped with a platform for evaluating ground reaction forces. (Free Med Base) during static (**A**), stabilographic (**B**) and dynamic (**C**) tests, along with sample reports from performed analyses.

**Table 1 ijerph-20-01652-t001:** Characteristics of the analysed groups, including people with Angle Class I, II and III.

Analysed Groups	N	Age [Years]			Body Height [cm]			Body Weight [kg]		
		Av.	Min.	Max.	Av.	Min.	Max.	Av.	Min.	Max.
I (Angle Class III)	30	22.77	20	30	170.93	157	186	68.23	48	100
II (Angle Class II)	30	23.87	19	35	172.8	157	183	66.93	48	91
III (Angle Class I)	30	22.63	19	32	171.73	155	192	67.27	47	90

**Table 2 ijerph-20-01652-t002:** Foot load distribution in groups with Angle Class I, II and III.

Parameters		Angle Class			*p*
		I Class	II Class	III Class	
Forefoot percentage weight-bearing left—LL [%]	Av ± SD	36.73 ± 10.35	45.53 ± 15.11	31.43 ± 14.26	*p* = 0.002 *
	Median	36	42.5	30	
	Quartiles	27.25–43.75	35.5–52	21–40.25	Cl.II > Cl.I, Cl.III
Forefoot percentage weight-bearing right—LL [%]	Av ± SD	37.1 ± 11.83	46.7 ± 14.96	29.57 ± 10.56	*p* < 0.001 *
	Median	38	46	28	
	Quartiles	29.25–44.5	35.25–57.75	21.25–36	Cl.II > Cl.I > Cl.III
Hindfoot percentage weight-bearing—left LL [%]	Av ± SD	63.27 ± 10.35	54.47 ± 15.11	67.07 ± 16.61	*p* = 0.004 *
	Median	64	57.5	70	
	Quartiles	56.25–72.75	48–64.5	57.5–79	Cl.III, Cl.I > Cl.II
Hindfoot percentage weight-bearing—left LL [%]	Av ± SD	62.9 ± 11.83	53.3 ± 14.96	68.6 ± 13.12	*p* < 0.001 *
	Median	62	54	71	
	Quartiles	55.5–70.75	42.25–64.75	61.75–74.75	Cl.III, Cl.I > Cl.II
Mean forefoot pressure values—left LL [g/cm^2^]	Av ± SD	295.73 ± 67.06	329.53 ± 63.6	283.47 ± 70.63	*p* = 0.018 *
	Median	295.5	329.5	290.5	
	Quartiles	245.5–329.5	303.75–355.5	247–323.25	Cl.II > Cl.I, Cl.III
Mean forefoot pressure values—right LL [g/cm^2^]	Av ± SD	284.57 ± 60.48	351.7 ± 62.8	293.67 ± 114.31	*p* < 0.001 *
	Median	288.5	346	267	
	Quartiles	233–318.25	307–386	237.5–313.25	Cl.II > Cl.III, Cl.I
Mean hindfoot pressure values—left LL [g/cm^2^]	Av ±SD	540.17 ± 120.86	488.03 ± 157.95	608.3 ± 263.09	*p* = 0.142
	Median	523	501.5	569	
	Quartiles	474.75–636.75	371–601	465.75–682.25	
Mean hindfoot pressure values—right LL [g/cm^2^]	Av ± SD	515.17 ± 116.6	460.47 ± 121.8	637.3 ± 316.44	*p* = 0.002 *
	Median	498.5	431.5	573	
	Quartiles	435.5–583	400–553.25	511–704	Cl.III > Cl.I, Cl.II

*p*—Kruskal–Wallis test + post hoc analysis (Dunn’s test). * Statistically significant difference. LL—Lower limb.

**Table 3 ijerph-20-01652-t003:** The Angle Class and CoP displacement relative to the centre of the support polygon.

Parameters		Angle Class			*p*
		I Class	II Class	III Class	
CoP displacement in sagittal plane	Forward	11 (36.67%)	20 (66.67%)	5 (16.67%)	*p* < 0.001 *
	Backward	19 (63.33%)	10 (33.33%)	25 (83.33%)	
CoP displacement in frontal plane	Non	6 (20.00%)	9 (30.00%)	4 (13.33%)	*p* = 0.419
	Left	19 (63.33%)	14 (46.67%)	17 (56.67%)	
	Right	5 (16.67%)	7 (23.33%)	9 (30.00%)	
Lateral CoP displacement [cm]	Av ± SD	1.46 ± 0.89	1.95 ± 1.03	2.38 ± 1.21	*p* = 0.009 *
	Median	1.12	2.01	1.97	
	Quartiles	0.8–2.21	1.16–2.22	1.41–3.4	Cl.III > Cl.I

*p*—for quantitative variables Kruskal–Wallis test + post hoc analysis (Dunn’s test), for qualitative variables chi-square test or Fisher’s exact test. * Statistically significant difference.

**Table 4 ijerph-20-01652-t004:** The results of selected stabilographic examination parameters.

Romberg Test (on both Feet, Eyes Open)		Angle Class			*p*
		I Class (N = 30)	II Class (N = 30)	III Class (N = 30)	
CoP path length [mm]	Av ± SD	502.81 ± 119.75	645.89 ± 236.91	748.86 ± 243.38	*p* < 0.001 *
	Median	496.74	576.86	689.7	
	Quartiles	403.48–573.34	482.06–772.48	595.54–855.47	Cl.III, Cl.II > Cl.I
Ellipse area [mm^2^]	Av ± SD	90.01 ± 75.1	166.81 ± 134.7	192.07 ± 161.17	*p* = 0.004 *
	Median	64.28	109.3	129.58	
	Quartiles	34.87–128.2	58.54–299.67	91.96–308.18	Cl.III, Cl.II > Cl.I
Mean CoP sway rate [mm/s]	Av ± SD	18.48 ± 5.04	22 ± 8.27	24.87 ± 9.46	*p* = 0.012 *
	Median	19.26	19.81	23.2	
	Quartiles	15.03–20.24	15.22–26.64	19.58–30.66	Cl.III > Cl.I
Romberg test (on both feet, eyes closed)		Angle Class			*p*
		I Class(N = 30)	II Class(N = 30)	III Class(N = 30)	
CoP path length [mm]	Av ± SD	552.53 ± 168.72	613.17 ± 206.2	745.91 ± 303.7	*p* = 0.018 *
	Median	517.33	561.62	723.22	
	Quartiles	432.88–609.83	455.52–734.96	511.29–905.92	Cl.III > Cl.I
Ellipse area [mm^2^]	Av ± SD	97.06 ± 110.42	159.65 ± 181.06	126.31 ± 88.62	*p* = 0.03 *
	Median	41.3	76.7	90.56	
	Quartiles	23.77–109.93	54.98–172.12	58.5–169.96	Cl.II, Cl.III > Cl.I
Mean CoP sway rate [mm/s]	Av ± SD	19.15 ± 6.13	21.03 ± 6.97	25.94 ± 10.5	*p* = 0.013 *
	Median	17.24	19.4	24.36	
	Quartiles	14.54–23.02	15.65–25.5	17.55–31.5	Cl.III > Cl.I

*p*—Kruskal–Wallis test + post hoc analysis (Dunn’s test). * Statistically significant difference.

**Table 5 ijerph-20-01652-t005:** Results of selected stabilographic examination parameters (one-legged stance test, eyes open) in groups with I, II and III Angle Class.

One-Legged Stance Test (Eyes Open)		Angle Class			*p*
		I Class (N = 30)	II Class (N = 30)	III Class (N = 30)	
CoP path length—left LL [mm]	Av ± SD	417.57 ± 105.61	461.88 ± 108.93	514.06 ± 164.36	*p* = 0.031 *
	Median	406.42	437.28	485.62	
	Quartiles	355.4–455.47	365.33–512.12	409.94–570.93	Cl.III > Cl.I
CoP path length—right LL [mm]	Av ± SD	486.98 ± 74.98	523.49 ± 137.94	543.5 ± 166.75	*p* = 0.659
	Median	478.93	483.74	488.61	
	Quartiles	431.15–545.38	415.97–598.14	430.94–597.96	
Mean CoP sway rate—left LL [mm/s]	Av ± SD	30.94 ± 10.03	36.29 ± 11.58	40.92 ± 13.63	*p* = 0.005 *
	Median	30.76	33.99	39.44	
	Quartiles	23.26–33.81	27.02–42.21	31.87–50.22	Cl.III > Cl.I
Mean CoP sway rate—right LL [mm/s]	Av ± SD	30.74 ± 7	34.15 ± 13.42	35.9 ± 13.78	*p* = 0.66
	Median	31.34	30.57	31.66	
	Quartiles	25.76–35.89	25.18–39.97	25.99–43.22	
Ellipse area—left LL [mm^2^]	Av ± SD	410.24 ± 313.45	428.3 ± 247.6	417.79 ± 236.48	*p* = 0.683
	Median	299.64	398.24	357.7	
	Quartiles	223.85–516.29	280.85–487.85	267.64–443.97	
Ellipse area—right LL [mm^2^]	Av ± SD	483.21 ± 317.8	681.3 ± 523.96	554.71 ± 378.34	*p* = 0.31
	Median	430.66	551.07	392.76	
	Quartiles	307.94–609	324.91–839.69	279.22–787.9	

*p*—Kruskal–Wallis test + post hoc analysis (Dunn’s test). * Statistically significant difference. LL—Lower limb.

**Table 6 ijerph-20-01652-t006:** Results of selected stabilographic examination parameters (one-legged stance test, eyes closed) in groups with I, II and III Angle Class.

One-Legged Stance Test (Eyes Closed)		Angle Class			*p*
		I Class (N = 30)	II Class (N = 30)	III Class (N = 30)	
CoP path length—left LL [mm]	Av ± SD	795.56 ± 129.87	902.89 ± 331.24	926.49 ± 441.19	*p* = 0.574
	Media	778.53	858.54	855.54	
	Quartiles	710.1–895.52	644.78–1018.92	702.32–962.43	
CoP path length—right LL [mm]	Av ± SD	783.5 ± 149.87	896.41 ± 224.2	920.13 ± 408.51	*p* = 0.161
	Median	771.24	896.14	820.1	
	Quartiles	685.24–876.58	742.9–1067.58	695.88–1071.5	
Mean CoP sway rate—left LL [mm/s]	Av ± SD	61.63 ± 12.27	71.08 ± 33.68	72.98 ± 30.36	*p* = 0.314
	Median	63.64	67.07	69.26	
	Quartiles	51.44–68.04	46.44–82.44	54.03–77.94	
Mean CoP sway rate -right LL [mm/s]	Av ± SD	60.85 ± 14.65	71.91 ± 22.12	75.45 ± 40.52	*p* = 0.125
	Median	57.51	70.38	61.44	
	Quartiles	50.62–69.31	56.76–87.89	54.03–89.28	
Ellipse area—left LL [mm^2^]	Av ± SD	2202.47 ± 1408.12	3078.82 ± 3387.25	2680.4 ± 2512.45	*p* = 0.878
	Median	1951.48	1812.28	1784.12	
	Quartiles	1377.34–2598.22	1250.5–3821.85	1125.59–2876.56	
Ellipse area—right LL [mm^2^]	Av ± SD	2199.2 ± 1163.97	3037.05 ± 1602.42	5722.58 ± 15289.18	*p* = 0.11
	Median	1819.48	3005.4	2545.61	
	Quartiles	1293.87–2904.57	1724.28–4004.04	1269.19–4068.25	

*p*—Kruskal–Wallis test + post hoc analysis (Dunn’s test). LL—Lower limb.

**Table 7 ijerph-20-01652-t007:** Romberg’s ratio values for parameters obtained in stabilographic examination.

Romberg Test (on Both Feet)		Angle Class			*p*
		I Class(N = 30)	II Class (N = 30)	III Class (N = 30)	
RR for CoP path length	Av ± SD	1.15 ± 0.43	0.98 ± 0.2	1.03 ± 0.37	*p* = 0.49
	Median	1.05	1.01	0.98	
	Quartiles	0.78–1.42	0.88–1.09	0.86–1.19	
RR for ellipse area	Av ± SD	1.9 ± 2.64	1.44 ± 2.38	0.89 ± 0.61	*p* = 0.88
	Median	0.98	0.95	0.66	
	Quartiles	0.26–1.99	0.39–1.42	0.47–1.22	
RR for mean CoP sway rate	Av ± SD	1.09 ± 0.43	0.99 ± 0.19	1.08 ± 0.34	*p* = 0.762
	Median	0.95	1	1	
	Quartiles	0.82–1.25	0.9–1.09	0.91–1.21	
One-legged stance test (left LL)		Angle Class			*p*
		I Class (N = 30)	II Class (N = 30)	III Class (N = 30)	
RR for CoP path length	Av ± SD	2 ± 0.51	1.97 ± 0.6	1.88 ± 0.72	*p* = 0.421
	Median	1.96	1.79	1.73	
	Quartiles	1.69–2.21	1.53–2.29	1.36–2.12	
RR for ellipse area	Av ± SD	7.69 ± 6.6	10.12 ± 13.23	7.79 ± 7.96	*p* = 0.778
	Median	7.5	4.84	4.6	
	Quartiles	3.43–9.31	3.56–8.67	3.46–8.31	
RR for mean CoP sway rate	Av ± SD	2.15 ± 0.71	2 ± 0.84	1.9 ± 0.8	*p* = 0.227
	Median	2.1	1.75	1.77	
	Quartiles	1.72–2.61	1.41–2.36	1.24–2.23	
One-legged stance test (right LL)		Angle Class			*p*
		I Class (N = 30)	II Class (N = 30)	III Class (N = 30)	
RR for CoP path length	Av ± SD	1.63 ± 0.32	1.75 ± 0.4	1.71 ± 0.49	*p* = 0.576
	Median	1.52	1.64	1.68	
	Quartiles	1.43–1.83	1.49–1.93	1.4–1.9	
RR for ellipse area	Av ± SD	6.58 ± 6.06	6.3 ± 4.46	18.38 ± 62.42	*p* = 0.861
	Median	4.65	4.88	4.72	
	Quartiles	2.52–8.35	2.74–8.98	2.92–11.53	
RR for mean CoP sway rate	Av ± SD	2.05 ± 0.56	2.26 ± 0.7	2.16 ± 0.76	*p* = 0.598
	Median	1.96	2.15	2.12	
	Quartiles	1.6–2.3	1.61–2.75	1.6–2.31	

*p*—Kruskal–Wallis test. RR—Romberg ratio. LL—lower limb.

**Table 8 ijerph-20-01652-t008:** Results of selected dynamic test parameters.

Parameters		Angle Class			*p*
		I Class	II Class	III Class	
Length of gait line—left LL [mm]	Av ± SD	220.13 ± 18.87	223.07 ± 15.38	219.03 ± 17.29	*p* = 0.596
	Median	217.5	220.5	218.5	
	Quartiles	204.75–234.75	213.25–234	211–230.75	
Length of gait line—right LL [mm]	Av ± SD	278.63 ± 327.49	220.6 ± 16.46	214.03 ± 22.45	*p* = 0.407
	Median	217	220	213	
	Quartiles	207.25–227.5	210.5–228.25	199.25–226.75	
Forefoot weight-bearing distribution— left LL [%]	Av ± SD	64.13 ± 5.18	63.07 ± 5.2	65.03 ± 4.73	*p* = 0.447
	Median	64.5	63	64	
	Quartiles	61.25–67	61–66	62–67	
Forefoot weight-bearing distribution— right LL [%]	Av ± SD	64.73 ± 5.24	63.63 ± 5.66	64.4 ± 5.27	*p* = 0.876
	Median	65.5	64.5	63.5	
	Quartiles	61.25–67.75	61–66.75	61–67	
Hindfoot weight-bearing distribution— left LL [%]	Av ± SD	35.87 ± 5.18	36.93 ± 5.2	34.97 ± 4.73	*p* = 0.447
	Median	35.5	37	36	
	Quartiles	33–38.75	34–39	33–38	
Hindfoot weight-bearing distribution— right LL [%]	Av ± SD	35.27 ± 5.24	36.37 ± 5.66	35.6 ± 5.27	*p* = 0.876
	Median	34.5	35.5	36.5	
	Quartiles	32.25–38.75	33.25–39	33–39	
Medial weight-bearing distribution— left LL [%]	Av ± SD	48.63 ± 6.29	50.97 ± 5.67	47.77 ± 7.47	*p* = 0.235
	Median	49.5	50.5	49	
	Quartiles	43.5–52.75	47–54	42.25–53.5	
Lateral weight-bearing distribution— left LL [%]	Av ± SD	51.37 ± 6.29	49.03 ± 5.61	52.23 ± 7.39	*p* = 0.193
	Median	50.5	49.5	51	
	Quartiles	47.25–56.5	46–53	48–57.75	
Medial weight-bearing distribution— right LL [%]	Av ± SD	49.7 ± 5.39	52.03 ± 6.09	49.97 ± 6.3	*p* = 0.315
	Median	49.5	52.5	51	
	Quartiles	45.25–54	48.25–55	45–54.75	
Lateral weight-bearing distribution— right LL [%]	Av ± SD	50.3 ± 5.39	47.97 ± 6.07	50.03 ± 6.29	*p* = 0.372
	Median	50.5	47.5	49	
	Quartiles	46–54.75	45–51.75	45.25–55	
Mean foot propulsion rate— left foot [mm/s]	Av ± SD	779.43 ± 128.84	773.31 ± 101.8	841.76 ± 355.66	*p* = 0.937
	Median	748.08	783.92	796.84	
	Quartiles	702.19–821.66	709.22–844.64	663.34–855.16	
Mean foot propulsion rate— right foot [mm/s]	Av ± SD	778.03 ± 179.68	792.8 ± 172.79	792.89 ± 120	*p* = 0.213
	Median	745.1	756.32	805.33	
	Quartiles	700.04–816.92	710.55–852.4	735.35–888.61	
Mean foot pressure— left foot [kg]	Av ± SD	45.16 ± 9.79	45.6 ± 9.27	46.76 ± 10.9	*p* = 0.844
	Median	45.22	43.12	46.44	
	Quartiles	37.88–50.2	40.01–51.88	39.89–50.13	
Mean foot pressure— right foot [kg]	Av ± SD	44.94 ± 9.93	45.42 ± 8.97	46.71 ± 10.91	*p* = 0.878
	Median	45.19	43.42	45.52	
	Quartiles	36.42–50.24	40.56–51.44	39.42–50.55	

*p*—Kruskal–Wallis test. RR—Romberg ratio. LL—Lower limb.

## Data Availability

Not applicable.

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
