# Peer review of "Is There a Correlation between Dental Occlusion, Postural Stability and Selected Gait Parameters in Adults?"

_ijerph, 2023, doi:10.3390/ijerph20021652_

Round 1

Reviewer 1 Report

Essentially: the topic is very interesting but the paper must be improved to be valued properly by the readers because there is a little bit of confusion.

Specifically: the purpose, the advantages and the limits of this research must be better specified. I think it's really worth it because the research topic and the results obtained are very interesting. Scholars of the subject expect further studies documenting the relationship between dental occlusion and body posture, a very hot topic since the nineties.

Suggestions:

Title: it is too confusing. Please change the too confusing title language, for example: Is there a correlation between dental occlusion and postural stability in adults?

Language: the language must be reviewed by a native speaker

ABSTRACT and INTRODUCTION: must be rewritten, the purpose needs to be better specified

Abstract: The purpose must be better specified

Introduction: it lacks fluidity and the purpose needs to be better specified

Material and methods: it is important to write down the ethics committee approval number

Conclusions: must be rewritten, are too schematic and any cases they do not highlight the usefulness of the study and the clinical relevance of the study

Reviewer 2 Report

Dear Authors,

thank you for giving me the opportunity to revise the manuscript entitled " Correlation between antero-posterior malocclusion, postural stability disorders and selected gait parameters in adults." The topic is interesting and can give important information in field. The manuscript is well written and succinct. Neverthless, there are some critical issues to address:

Ø  Introduction: In my opinion, the introduction should be improved in therapeutic approach of postural stability disorders and its correlation with TMD. Please, read “Ferrillo M, Marotta N, Giudice A, Calafiore D, Curci C, Fortunato L, Ammendolia A, de Sire A. Effects of Occlusal Splints on Spinal Posture in Patients with Temporomandibular Disorders: A Systematic Review. Healthcare (Basel). 2022 Apr 15;10(4):739. doi: 10.3390/healthcare10040739.”

·        Please, explain in other way the gol of the study. This sentence is quite confuse

Ø  Material and methods:

·        Please, move the number and age of patients in results.

·        Have you data about more detailed demographic characteristics of the population?

Ø  Results: Well done

Ø  Discussion: Well done

Best Regards

Round 2

Reviewer 2 Report

Dear Authors,

at the light of my knowledge and after your outstanding revision work, the paper is suitable fo fully publication in Journal.

Best Regards